# Partitioning Structure Learning for Segmented Linear Regression Trees

**Xiangyu Zheng**
Peking University
zhengxiangyu@pku.edu.cn

**Song Xi Chen**
Peking University
csx@gsm.pku.edu.cn

## Abstract

This paper proposes a partitioning structure learning method for segmented linear regression trees (SLRT), which assigns linear predictors over the terminal nodes. The recursive partitioning process is driven by an adaptive split selection algorithm that maximizes, at each node, a criterion function based on a conditional Kendall's $\tau$ statistic that measures the rank dependence between the regressors and the fitted linear residuals. Theoretical analysis shows that the split selection algorithm permits consistent identification and estimation of the unknown segments. A sufficiently large tree is induced by applying the split selection algorithm recursively. Then the minimal cost-complexity tree pruning procedure is applied to attain the right-sized tree, that ensures (i) the nested structure of pruned subtrees and (ii) consistent estimation to the number of segments. Implanting the SLRT as a built-in base predictor, we obtain the ensemble predictors by random forests (RF) and the proposed weighted random forests (WRF). The practical performance of the SLRT and its ensemble versions are evaluated via numerical simulations and empirical studies. The latter shows their advantageous predictive performance over a set of state-of-the-art tree-based models on well-studied public datasets.

## 1   Introduction

Data partitioning is a fundamental pre-processing method that explores the partitioning structure of the feature space such that the subspaces are more compliant to a simple model [1]. We consider the segmented linear regression (SLR) models, which prescribes linear predictors over the partitions. Partitioning structure learning is the core of SLR, that selects the split variables and levels as well as determines the number of segments.

SLR has been studied in statistics and econometrics [2, 3, 4, 5], but the existing methods tend to assume the split variable is known and univariate, with segments estimated by a costly simultaneous optimization. We propose a tree-based approach for SLR called segmented linear regression trees (SLRT), that does not require the pre-specified information about the split variables. SLRT is completely data-driven and facilitates more efficient computation via recursive partitioning, which is fundamentally based on a split selection algorithm and a tree pruning algorithm.

**Split Selection Algorithm** At each internal node of the tree, the optimal split variable and level pair is selected to partition the feature space into two halves. Let $\hat{e}$ be the fitted residuals by the ordinary least square regression. Any non-linearity in the underlying regression function is reflected in the dependence between $\hat{e}$ and the regressors. Based on the conditional Kendall's $\tau$ rank correlation [6], we propose the following criterion function at a candidate split variable index $j$ and a split level $a$, $\mathcal{C}(j,a) = \sum_{k=1}^{p} \{|\hat{\tau}(X_k, \hat{e}|X_j \leq a)| + |\hat{\tau}(X_k, \hat{e}|X_j > a)|\}$, where $\hat{\tau}$ is the sample version of the Kendall's $\tau$, $X$ is a $p$-dimensional regressors vector with $X_k$ being its $k$-th component. The optimal split is selected by maximizing $\mathcal{C}(j,a)$ over the candidate split variables $\{X_j\}$ and levels $\{a\}$ in the

observed sample of $X_j$. Theoretical analysis shows that it leads to the consistent identification and estimation of the most prevailing split variable and level that attains the maximum of $\mathcal{C}(j, a)$.

**Tree Pruning Algorithm** We define an adaptive cost-complexity measure that combines the accuracy of the linear regression fit at each node with a penalty for a large tree size. The optimally pruned tree is selected from a nested sequence of pruned subtrees by minimizing the cost-complexity measure. Theoretical analysis shows that the pruning method leads to consistent estimation of the underlying number of segments, which promotes a parsimonious partitioning structure.

**Leaf Modeling and Ensemble Methods** For predictors within segments, we employ the LASSO procedure [7] to select the most influential variables and estimate the linear parameters. Furthermore, by implanting SLRT as the base predictor in the random forests (RF) formulation, we obtain the ensemble predictor that improves the model stability and predictive accuracy. A weighted version of the RF (WRF) is also proposed, which shows an improved performance over the RF by reducing the importance of those under-performing trees in weighting.

As a novel tree-based learning method for segmented linear models, SLRT possesses attractive theoretical properties of the consistent identification and estimation of the partitioning structures, which are confirmed favorably in numerical simulations. Applied on nine benchmark datasets, SLRT had advantageous predictive performance over several state-of-the-art tree-based methods, with further improvement offered by the RF and WRF with SLRT as the base predictor.

The source code of the algorithm is available in the supplementary material.

## 1.1 Related Work

The proposed segmented linear regression tree is a tree-based approach to segmented linear regression (SLR) models, where the partitions of the feature space is axis-aligned. The existing methods of SLR tend to assume a known split variable, such that the partitioning structure learning is reduced to the change-points detection with respect to a given variable. For instance, [2, 3] considered the case where both the univariate partitioning variable and the number of segments are pre-specified. [4, 5] estimated the number of change-points by minimizing the Bayesian information criteria (BIC). [8] selected the change-points via the sum of squared residuals in conjunction with the permutation test, which also assumed a known split variable. Our approach does not require pre-specified information of the segments, and learns the partitioning structure via a tree induction process.

SLR also belongs to the class of region-specific linear models. [1] proposed a partition-wise linear model, where axis-aligned partitions are pre-specified and an 0-1 activeness function was assigned to each region. With each region-specific linear model being estimated first, the activeness functions are optimized through a global convex loss function. [9] proposed a local supervised learning through space partitioning for classification which allows arbitrary partitions and considered linear classifiers, while [10] employed a Bayesian updating process to partition the feature space to rectangular bounding boxes and assigned a constant estimation over each partition like CART.

Our approach is closely related to the regression tree (regression part of CART, [11]), a well-known region-specific approach that used a constant-valued predictor within each terminal node. There have been tree-based algorithms which assigns linear predictors in terminal nodes, which tend to be heuristic without theoretical analysis. One group of the methods [12, 13, 14] adopted splitting algorithms similar to that of CART, which tend to ignore the correspondence between the evaluation criteria for splits and the models in terminal nodes. Another group [15, 16, 17, 18, 19] employed heuristic criteria designed to make the subsets more compliant for linear models in one step, without considering the properties of the estimated boundaries. Our split selection algorithm is closely related to GUIDE [17, 18] as both utilize the estimated residuals $\hat{e}$ at a node level. However, GUIDE used the signs of $\hat{e}$ that would be less informative than using $\hat{e}$ via the Kenall's $\tau$. Another difference is that GUIDE considered the marginal association between signs of $\hat{e}$ and the regressors instead of conditioning on a split variable and level, which can lead to the mis-identification of the split variable.

## 2 Segmented Linear Regression Models

This section presents the framework of SLRT, and provides the motivation and the theoretical properties to the computational algorithms in Section 3.

## 2.1 Framework

Consider the relationship between a univariate response $Y$ and a multivariate explanatory covariate $X = (X_1, \cdots, X_p)^T$. Assume that the mean regression function $m(X) = E(Y|X)$ is partition-wise linear over $L_0$ unknown partitions $\{D_l\}_{l=1}^{L_0}$ in the domain of $X$ so that

$$Y = \sum_{l=1}^{L_0} (\alpha_l + X'\beta_l)\, \mathbb{1}(X \in D_l) + \varepsilon, \tag{1}$$

where $(\alpha_l, \beta_l)$ are regression coefficients over domain $D_l$ and $\varepsilon$ is the random error satisfying $E(\varepsilon|X) = 0$. In this paper, we consider the case of axis-aligned partitions $\{D_l\}$, which are determined by a collection of split levels $\{X_{j_q} = a_q\}_{q=1}^{Q_0}$. The model may be extended to more general shape of $\{D_l\}$ by undergoing pre-transformations, which will be a topic for a future study.

The determination of $\{D_l\}$ is equivalent to selecting the split variable and level pairs, namely $\mathcal{S} = \{(j_q, a_q)\}_{q=1}^{Q_0}$, where $Q_0$ is determined by $L_0$ and the geometry of $\{D_l\}$. The task of partitioning structure learning is to identify the underlying split variables and estimate the split levels consistently. We adopt the computationally efficient regression tree approach by applying the split variable and level selection algorithm recursively, ending with terminal nodes for the desired partitions.

## 2.2 Statistical Analysis of Criterion Functions

To select the optimal split variable and level at a node, we fit the least square regression over the node and select the optimal split by studying the rank correlation between the estimated residuals and the regressors given a candidate split variable and a split level. This is computationally more efficient than the commonly used cost minimization procedure [11, 17, 20], which would require repeated least square fitting for each candidate split variable and level.

For the ease of presentation, we consider the one-time split selection over the root node $t_0$ which contains data $D_{t_0}$ of $n$ independent observations $\{X(i), Y(i)\}_{i=1}^n$ generated from Model (1). We are to partition $D_{t_0}$ into two subsets to make the data on each subset more compliant to a linear model. To attain this, let $\hat{Y} = \hat{\alpha}_{t_0} + X'\hat{\beta}_{t_0}$ be the fitted ordinary least square (OLS) regression over $D_{t_0}$, and $\hat{e} = Y - \hat{Y}$ be the estimated residuals. If the underlying regression function $m(X)$ is nonlinear, the non-linearity will be reflected in the residuals $\hat{e}$ and their dependence with the potential split variables. Indeed, if $m(X)$ is piecewise linear, the estimated residuals $\hat{e}$ is also piecewise linear in $X$ since $\hat{e} = \sum_{l=1}^{L} \left( (\alpha_l - \hat{\alpha}_{t_0}) + X'(\beta_l - \hat{\beta}_{t_0}) \right) \mathbb{I}(X \in D_l) + \varepsilon$. A regressor $X_k$ and $\hat{e}$ tend to be accordant (discordant) for positive (negative) coefficient within each partition. To capture the dependence, we employ the Kendall's $\tau$ coefficient [6] to define the following criterion function:

$$\mathcal{C}(j, a) = \sum_{k=1}^{p} \left\{ |\hat{\tau}(X_k, \hat{e}|X_j \leq a)| + |\hat{\tau}(X_k, \hat{e}|X_j > a)| \right\}, \tag{2}$$

for $1 \leq j \leq p$, $a \in \{X_j(i)\}_{i=1}^n$ and

$$\hat{\tau}(X_k, \hat{e}|X_j \leq a) = \frac{\displaystyle\sum_{\substack{i, i' \in I_{t_L}(j,a) \\ i < i'}} \operatorname{sgn}\left((X_k(i) - \hat{e}(i))(X_k(i') - \hat{e}(i'))\right)}{N_{t_L}(j, a)(N_{t_L}(j, a) - 1)/2} \tag{3}$$

is the Kendall's $\tau$ statistic, where $I_{t_L}(j, a) = \{i \,|\, X_j(i) \leq a, 1 \leq i \leq n\}$ is the index set for the left partition split by variable $X_j$ at level $a$ and the sample size of $I_{t_L}(j, a)$ is $N_{t_L}(j, a) = |I_{t_L}(j, a)|$. The $\hat{\tau}(X_k, \hat{e}|X_j > a)$, $I_{t_R}(j, a)$ and $N_{t_R}(j, a)$ are defined analogously.

Based on $\mathcal{C}(j, a)$, we propose the split selection Algorithm 1 in Section 3.1, which is essentially motivated by the following Lemma 2.1.

**Lemma 2.1** *Suppose the regressors are uncorrelated conditional on each partition of $\{D_l\}_{l=1}^{L_0}$. Assume the following technical conditions: (1) $\mathrm{E}(\varepsilon|X) = \mathrm{median}(\varepsilon|X) = \mathrm{median}(X_j - \mathrm{E}(X_j)|X_k) = 0$ for $k \neq j$; (2) for $(j_k, a_k) \in \mathcal{S}$, $0 < \mathrm{P}(X_{j_k}^{(s)} \leq a_k | X_{j'}^{(s)} \leq a') < 1$ when $(j', a') \neq (j_k, a_k)$. Let $\bar{\mathcal{C}}(j, a)$ be the probability limit of $\mathcal{C}(j, a)$. Then, for any $(j', a') \notin \mathcal{S}$, $\bar{\mathcal{C}}(j', a') < \bar{\mathcal{C}}(j_q, a_q)$ for any $(j_q, a_q) \in \mathcal{S}$, with $\mathcal{S}$ being the genuine set of split variable and level pairs.*

The proof of Lemma 2.1 includes two phases. We firstly investigate the simple case where $L_0 = 2$, and then generalize the conclusion to $L_0 \geq 2$ using the law of iterated expectation. Please refer to the supplements for the details and a further discussion about the technical conditions. Intuitively speaking, maximizing $\mathcal{C}(j, a)$ is to maximize the sum of rank correlations between the estimated residuals $\hat{e}$ and each element of the regressors $X$ over each of the selected subsets ($\{X_j \leq a\}$ and $\{X_j > a\}$), such that the rank correlation with $X$ contained in $\hat{e}$ could be further distilled by regressing $\hat{e}$ on the regressors conditional on each subset, which leads to a segmented linear regression.

Define the distance $d((\hat{j}, \hat{a}), \mathcal{S}) = \min_q \{|(\hat{j}, \hat{a}) - (j_q, a_q)| \big| (j_q, a_q) \in \mathcal{S}\}$. Then, Lemma 2.1 leads to the following theorem that validates the consistency property of the selected split.

**Theorem 2.1** *Let* $(\hat{j}, \hat{a}) = \operatorname{argmax} \mathcal{C}(j, a)$. *Then,* $\mathrm{P}\left(d((\hat{j}, \hat{a}), \mathcal{S}) > \varepsilon\right) \to 0$ *as* $n \to \infty$ *under the assumptions of Lemma 2.1. Specially, when* $\bar{\mathcal{C}}(j, a)$ *has a unique maximum* $(j^*, a^*)$, *we have* $(j^*, a^*) \in \mathcal{S}$ *and* $(\hat{j}, \hat{a}) \xrightarrow{\mathrm{P}} (j^*, a^*)$ *as* $n \to \infty$.

When the regressors are not conditional uncorrelated as required by Lemma 2.1, we conduct a linear transformation when calculating the conditional Kendall's $\tau$ coefficients with $\hat{e}$. Specifically, let $\mathbf{X} = (X(1), \cdots, X(n))'$ be the data matrix for $n$ observations of $X$. Given a split variable and level $X_j = a$, define $\mathbf{X}_L = \mathbf{X} \operatorname{diag}\{\mathbb{1}(X_j(1) \leq a), \cdots, \mathbb{1}(X_j(n) \leq a)\}$ and $\mathbf{X}_R = \mathbf{X} \operatorname{diag}\{\mathbb{1}(X_j(1) > a), \cdots, \mathbb{1}(X_j(n) > a)\}$. Then, there exists a non-singular matrix $P^{(j,a)}$ such that $\mathbf{Z}'\mathbf{Z}$ is diagonal for $\mathbf{Z} = \mathbf{X}P^{(j,a)}$, $\mathbf{X}_L P^{(j,a)}$ and $\mathbf{X}_R P^{(j,a)}$, which is facilitated by the simultaneous diagonalization of positive definite matrices (see supplements for detailed calculation procedures of $P^{(j,a)}$ that are based on the spectral decomposition). Let $Z^{(\bar{j},\bar{a})} = XP^{(\bar{j},\bar{a})}$ be the transformed regressors with $Z_k^{(\bar{j},\bar{a})}$ being the $k$-th element. Define the modified criterion function with index $(\bar{j}, \bar{a})$,

$$\mathcal{C}_{(\bar{j},\bar{a})}(j, a) = \sum_{k=1}^{p} \left\{ \left| \hat{\tau}\left( Z_k^{(\bar{j},\bar{a})}, \hat{e} | X_j \leq a \right) \right| + \left| \hat{\tau}\left( Z_k^{(\bar{j},\bar{a})}, \hat{e} | X_j > a \right) \right| \right\}, \qquad (4)$$

where $Z^{(\bar{j},\bar{a})}$ replaces $X$ in (2) and $\hat{e}$ is still the residuals of OLS regression at the node $t_0$ without transformation. The following Lemma 2.2 shows that $\mathcal{C}_{(\bar{j},\bar{a})}$ possesses properties similar to that of $\mathcal{C}$ introduced in Lemma 2.1, while Lemma 2.2 does not require $X$ to be conditional uncorrelated. This motivates the Algorithm 2 in Section 3.1 and leads to the convergence result in Theorem 2.2.

**Lemma 2.2** *Let* $\bar{\mathcal{C}}_{(\bar{j},\bar{a})}(j, a)$ *be the probability limit of* $\mathcal{C}_{(\bar{j},\bar{a})}(j, a)$, *Then, under the same technical conditions (1) and (2) as required in Lemma 2.1,* $\operatorname{argmax}_{(j,a)} \bar{\mathcal{C}}_{(\bar{j},\bar{a})}(j, a) = (\bar{j}, \bar{a})$ *when* $(\bar{j}, \bar{a}) \in \mathcal{S}$, *with* $\mathcal{S} = \{(j_q, a_q)\}_{q=1}^{Q_0}$ *being the genuine set of split variable and level pairs.*

**Theorem 2.2** *Let* $(\hat{j}, \hat{a})_{(\bar{j}, \bar{a})} = \operatorname{argmax}_{(j,a)} \mathcal{C}_{(\bar{j},\bar{a})}(j, a)$. *Under the technical conditions in Lemma 2.2, if* $(\bar{j}, \bar{a}) \in \mathcal{S}$, *then* $(\hat{j}, \hat{a})_{(\bar{j}, \bar{a})} \xrightarrow{\mathrm{P}} (\bar{j}, \bar{a})$ *as* $n \to \infty$.

Theorem 2.2 implies that the convergence of $(\hat{j}, \hat{a})_{(\bar{j},\bar{a})}$ to $(\bar{j}, \bar{a})$ is a necessary condition for $(\bar{j}, \bar{a}) \in \mathcal{S}$. This motivates the distance minimization procedure in Line 7 of Algorithm 2 .

## 3 Partitioning Structure Learning

We first use the recursive partitioning procedures to generate the initial partitions. Then, we employ the cost-complexity tree pruning procedure to obtain a parsimonious partitions structure.

### 3.1 Initial Partitions by Recursive Partitioning

The recursive partitioning needs a split selection algorithm at the node level, and a stopping rule for the termination of the partitioning process, the latter is based on two tuning parameters: $N_{\min}$ that controls the sample size in any leaf node and $\mathrm{Dep}_{\max}$ that limits the depth of the tree.

The split selection is the core of the recursive partitioning. In the following Algorithm 1 of recursive partitioning, the split is selected by maximizing $\mathcal{C}(j, a)$. This is directly motivated by Lemma 2.1,

that shows the maximum of $\bar{\mathcal{C}}(j,a)$ is within the genuine set of split variable and level pairs $\mathcal{S}$. Besides, according to Theorem 2.1, the selected split level $X_{\hat{j}} = \hat{a}$ determined by the maximum of the criterion function $\mathcal{C}(j,a)$ is consistent to one of the underlying genuine partitioning boundary provided the regressors are uncorrelated conditional on each partition.

---

**Algorithm 1** Recursive Partitioning for Conditional Uncorrelated Regressors

---

**Input:** Training data $D_{t_0} = \{(X_i, Y_i)\}_{i=1}^n$.
**Output:** Data partitions $\mathcal{D} = \{D_i\}_{i=1}^L$.
1: Initialize: No pre-specified partitions, $\mathcal{D} = \varnothing$; the depth of the root node $\mathrm{Dep}(t_0) = 0$.
2: **if** $|D_{t_0}| < 2N_{\min}$ or $\mathrm{Dep}(t_0) > \mathrm{Dep}_{\max}$ **then**
3:     **return** $\mathcal{D} = \mathcal{D} \cup \{D_{t_0}\}$
4: **else**
5:     Fit a least square linear regression of $Y$ on $X$ over $D_{t_0}$ and get the estimated residuals $\hat{e}$.
6:     Calculate the criterion function $\mathcal{C}(j,a)$ for $(j,a)$ in the set of candidate split pairs $C_{t_0} = \{(j,a) | j = 1, \cdots, p, a \in \{X_j(i) | (X(i), Y(i)) \in D_{t_0}\}, N_{\min} \le |\{i | X_j(i) > a\}| < |D_{t_0}| - N_{\min}\}$.
7:     $(\hat{j}, \hat{a}) = \mathrm{argmax}_{(j,a)} \mathcal{C}(j,a)$
8:     Let $t_L$ and $t_R$ be the left and right child-nodes of $t_0$, $\mathrm{Dep}(t_L) = \mathrm{Dep}(t_R) = \mathrm{Dep}(t_0) + 1$, with $D_{t_L} = \{(X(i), Y(i)) | X_{\hat{j}}(i) \le \hat{a}\} \cap D_{t_0}$ and $D_{t_R} = \{(X(i), Y(i)) | X_{\hat{j}}(i) > \hat{a}\} \cap D_{t_0}$.
9:     $t_0 \Leftarrow t_L$ and execute step $2 - 11$.
10:    $t_0 \Leftarrow t_R$ and execute step $2 - 11$.
11: **end if**

---

When taking the correlation between regressors into consideration, we apply Algorithm 2 to select the optimal split over the original untransformed variables, which retains easy interpretability of the partitions but requires higher computation cost as is analyzed in the following. Since Algorithm 1 has outlined the recursive partitioning process, Algorithm 2 will concentrate on the split selection at the node level, which corresponds to Line $5-7$ of Algorithm 1.

Enlightened by Theorem 2.2, we select the optimal split by minimizing the distance between $(\hat{j}, \hat{a})_{(\bar{j}, \bar{a})}$ and $(\bar{j}, \bar{a})$ in Algorithm 2, where the standardized distance $d(\hat{a}_{(\bar{j}, \bar{a})}, \bar{a}) = |\hat{a}_{(\bar{j}, \bar{a})} - \bar{a}| / \hat{\sigma}(X_{\bar{j}})$ is used for $\bar{j} = \hat{j}_{(\bar{j}, \bar{a})}$, with $\hat{\sigma}(X_{\bar{j}})$ being the sample standard deviation of $X_{\bar{j}}$.

---

**Algorithm 2** Split Selection for Correlated Regressors

---

**Input:** Training data $D_{t_0} = \{(X_i, Y_i)\}_{i=1}^n$, $|D_{t_0}| > 2N_{\min}$.
**Output:** The optimal split variable and level pair $(\hat{j}, \hat{a})$; or no splits and $t_0$ is a terminal node.
1: Fit a least square linear regression of $Y$ on $X$ over $D_{t_0}$ and get the estimated residuals $\hat{e}$.
2: **for** each $(\bar{j}, \bar{a}) \in C_{t_0}$ **do**
3:     calculate the criterion function $\mathcal{C}_{(\bar{j}, \bar{a})}(j,a)$ for each $(j,a) \in C_{t_0}$;
4:     $(\hat{j}, \hat{a})_{(\bar{j}, \bar{a})} = \mathrm{argmax}_{(j,a)} \mathcal{C}_{\bar{j}, \bar{a}}(j,a)$, the 'local' optimal split under $(\bar{j}, \bar{a})$,
5: **end for**
6: **if** $\{(\hat{j}, \hat{a})_{(\bar{j}, \bar{a})} | \bar{j} = \hat{j}_{(\bar{j}, \bar{a})}\} \ne \varnothing$ **then**
7:     **return** the optimal split $(\hat{j}, \hat{a}) = \mathrm{argmin}_{(\bar{j}, \bar{a})} \{d(\hat{a}_{(\bar{j}, \bar{a})}, \bar{a}) | (\bar{j}, \bar{a}) \in C_{t_0}, \bar{j} = \hat{j}_{(\bar{j}, \bar{a})}\}$.
8: **else**
9:     **return** no suitable splits, $t_0$ is a terminal node.
10: **end if**

---

As for the computation complexity of Algorithm 2, suppose there are $M_t$ candidate splits in a node $t$, then it involves $M_t^2$ times of calculations of the criterion functions. Since the calculation of Kendall's $\tau$ in $\mathcal{C}_{(\bar{j}, \bar{a})}(\cdot, \cdot)$ is of complexity $O(N_t \log(N_t))$, with $N_t$ being the sample size of node $t$. Hence the complexity of Algorithm 2 is $M_t^2 O(N_t \log(N_t))$, which is costly compared to Algorithm 1 that is only $M_t O(N_t \log(N_t))$. Therefore, we may adopt a stopping strategy that terminates the split process when the $\min d(\hat{a}_{(\bar{j}, \bar{a})}, \bar{a})$ in Line 7 in Algorithm 2 is larger than a given threshold.

Applying either Algorithm 1 or Algorithm 2 recursively for split selections leads to an initial tree $T_{\max}$, which determines the initial partitions. We outline the pruning of $T_{\max}$ in the following.

## 3.2 Minimal Cost-complexity Tree Pruning

We adopt the minimal cost-complexity pruning procedure in CART [11], but with a newly defined cost-complexity measure $I_\alpha(T)$ for the regression tree $T$ with linear regression models on leaves.

Define the accuracy measure at a node $t$ in a tree $T$ as $I(t) = \sum_{(X(i),Y(i))\in t}(Y(i) - \hat{m}_T(X_i))^2$, where $\hat{m}_T(\cdot)$ is the segmented liner regression function determined by $T$. The accuracy of $T$ is $I(T) = \sum_{t\in\tilde{T}} I(t)$, where $\tilde{T}$ denotes the set of leaf nodes in $T$ and $n$ is the sample size of the training data. The model complexity of $T$ is measured by the number of leaf nodes $|\tilde{T}|$.

Taking both the accuracy measure and model complexity into consideration, the cost-complexity measure $I_\alpha(T) = I(T)/n + \alpha|\tilde{T}|$, where $\alpha$ is a positive penalizing parameter. The optimally pruned tree $T(\alpha)$ is defined as the smallest subtree of $T_{\max}$ that minimizes $I_\alpha(T)$, same as the Definition 3.6 in [11]. Proposition 3.1 verifies the existence and the uniqueness of $T(\alpha)$ and the nested structure of $\{T(\alpha), \alpha > 0\}$ as $\alpha$ varies, which is essential for an efficient programming. The proof is by induction where the key is an inequality satisfied by $I_\alpha(T)$. Please refer to the supplementary for details.

**Proposition 3.1** *Let $T_{\max}$ be the initial tree, then*

    *(i) given an $\alpha$, there exists one optimally pruned subtree $T(\alpha)$ of $T_{\max}$;*

    *(ii) if $\alpha_2 > \alpha_1$, then $T(\alpha_2)$ is a subtree of or equal to $T(\alpha_1)$.*

To obtain the optimally pruned tree, the optimal complexity parameter $\alpha^*$ should be selected. Although $\alpha$ runs through a continuum of values, there are finite number of subtrees $T(\alpha)$, say $K$ subtrees of $T_{\max}$. Then by Proposition 3.1, there exists an increasing sequence of $\{\alpha_k | k = 1, \cdots, K\}$ such that $T(\alpha_{k+1}) \subset T(\alpha_k)$, and for $\alpha \in [\alpha_k, \alpha_{k+1})$, $T(\alpha) = T(\alpha_k)$. In fact, $\{\alpha_k\}_{k=1}^K$ can be exactly calculated from $T_{\max}$. Specifically, let $T_t$ be the subbranch of a tree $T$ with node $t$ being its root, then $\alpha_k = \min_t \{ \frac{I(t)-I(T_t)}{|\tilde{T}_t|-1} \big| t \in T \text{ and } t \notin \tilde{T} \}$ for $T = T_{\max}$ when $k = 1$ and $T = T(\alpha_{k-1})$ when $k > 1$.

Therefore, the optimization of $\alpha^*$ is reduced to selecting an optimal $k^*$ from $\{1, \cdots, K\}$. let $\bar{\alpha}_k = \sqrt{\alpha_k \alpha_{k+1}}$ for $1 \le k \le (K-1)$ and $\bar{\alpha}_K = \alpha_K$. The optimal complexity parameter $\alpha^*$ is selected from $\{\bar{\alpha}_k\}_{k=1}^K$ by the ten-fold cross-validation to optimize the average predictive accuracy measured by the sum of squared residuals. Then, the optimally pruned subtree is $T(\alpha^*)$. Let $\hat{L}$ be the number of terminal nodes in $T(\alpha^*)$, under certain general conditions for the distribution of $\varepsilon$ and given appropriate $\alpha^*$, it can be proved that $\hat{L}$ converges to the genuine number of segments $L_0$ in probability.

# 4 Leaf Modeling and Ensemble Methods

## 4.1 LASSO Linear Regression on Leaf Nodes

Let $\{\hat{D}_l\}_{l=1}^{\hat{L}}$ be the partitions structure determined by $T(\alpha^*)$. Confined on each partition, the regression coefficients $(\alpha_l, \beta_l)$ can be estimated by the ordinary least square. However, as $X$ is the overall regressors, not each of them necessarily owns a non-zero coefficient over $\hat{D}_l$, and the significant variables set within each $\hat{D}_l$ may vary. Thus, we consider the variables selection within each leaf node. Besides, as the partitioning process decreases the sample size for estimation in each node, we would like to determine a smaller set of variables that exhibit the strongest effects on each $\hat{D}_l$.

To this purpose, the LASSO method [7] is employed for the variables selection, where the regression coefficients $(\alpha_l, \beta_l)$ is estimated by

$$\hat{\alpha}_l^{\text{lasso}} = \bar{Y}_{\hat{D}_l}; \; \hat{\beta}_l^{\text{lasso}} = \underset{\beta_l}{\operatorname{argmin}} \sum_{\{i|X^{(s)}(i)\in\hat{D}_l\}} \{(Y(i) - \bar{Y} - X^{(r)}(i)'\beta_l)^2 + \lambda_l \sum_{j=1}^p |\beta_l(j)|\},$$

where $\bar{Y}_{\hat{D}_l}$ is the sample average over $\hat{D}_l$ and $\lambda_l$ is the shrinkage parameter selected by cross-validation.

Then, the final prediction function is $\hat{m}_{T(\alpha^*)}^{\text{lasso}}(X) = \sum_{l=1}^{\hat{L}} \left( \hat{\alpha}_l^{\text{lasso}} + X^{(r)'}\hat{\beta}_l^{\text{lasso}} \right) \mathbb{1}(X^{(s)} \in \hat{D}_l).$

## 4.2 Weighted Random Forests

We can implant the Kendall's $\tau$ based partitioning learning algorithm in random forests (RF, [21]) to create the ensemble predictor. Here we propose the weighted random forests (WRF), that considers the accuracy of each tree and puts the final predictor as a weighted average to improve the predictions.

Suppose $\{T_b\}_{b=1}^B$ are the $B$ regression trees induced from the bootstrap training sets. The RF takes the simple average over the predictions of all regression trees, that is, $\hat{m}_{\text{RF}}(X) = \frac{1}{B}\sum_{b=1}^B \hat{m}_{T_b}(X)$. Note given the training set and $X$, $\{\hat{m}_{T_1}(X), \cdots, \hat{m}_{T_B}(X)\}$ are independent random variables. The variance of $\hat{m}_{T_b}(X)$ reflects the predictive accuracy of regression tree $T_b$, that can be estimated by $I(t_{T_b}(X))$ for $t_{T_b}(X)$ is the leaf node in $T_b$ containing $X$. According to Proposition 4.1, taking the variances of single predictors into account will improve the accuracy of the ensemble predictor.

**Proposition 4.1** *Let $\{Z_b\}_{b=1}^B$ be independently distributed random variables from a population $P \in \mathcal{P}$ having a common mean $\mu$ and $\mathrm{Var}(Z_b) = \sigma_b^2$. Let $\mathcal{A}$ be the class of all unbiased linear estimations for $\mu$. Then, the optimal estimation that minimizes $\mathrm{E}(A - \mu)^2 (A \in \mathcal{A})$ is $\sum_{b=1}^B \frac{1/\sigma_b^2}{\sum_{j=1}^B 1/\sigma_j^2} Z_b$.*

Motivated by Proposition 4.1, we propose the weighted random forests (WRF), which shows improved predictive accuracy over the RF on the benchmark datasets in Section 5.2.

$$\hat{m}_{\text{WRF}}(X) = \sum_{b=1}^B \frac{1/I(t_{T_b}(X))}{\sum_{j=1}^B 1/I(t_{T_j}(X))} \hat{m}_{T_b}(X). \tag{5}$$

# 5 Experimental Results

## 5.1 Simulation Study

In this part, we would illustrate SLRT with two examples. One is segmented linear regression function to investigate the performance of partitions structure learning, the other is a general continuous regression function considered in [22], to illustrate how our method works under a general setting.

First, we consider a regression function $m(X)$ that is segmented linear with 12 segments determined by 4 binary splits at $X_1 = 10, X_2 = 10, X_2 = 15, X_4 \in \{a, b\}$ or $\{c\}$:

$$m(X) = 3X_1\mathbb{I}\{X_2 > 15\} - 3X_1\mathbb{I}\{X_2 \leq 15\} - 3X_2\mathbb{I}\{X_2 > 10\} - 5X_2\mathbb{I}\{X_2 \leq 10\}$$
$$+ X_3\mathbb{I}\{X_1 > 10\} - X_3\mathbb{I}\{X_1 \leq 10\} + X_3\mathbb{I}\{X_4 \in \{\text{`a', `b'}\}\} - 3X_3\mathbb{I}\{X_4 \in \{\text{`c'}\}\}. \tag{6}$$

Training data of size 1500 was generated from $Y = m(X) + \varepsilon$ with $\varepsilon \sim N(0, 1)$ and independent regressors $X_1 \sim U(0, 20)$, $X_2 \sim U(0, 25)$, $X_3 \sim U(0, 10)$ and $X_4$ took values in $\{\text{`a', `b', `c'}\}$ with equal probabilities (see Supplementary for the case of dependent regressors). Figure 1 shows that the estimated partitions in the terminal nodes of $T(\alpha^*)$ are quite close to the space partitions in $m(X)$.

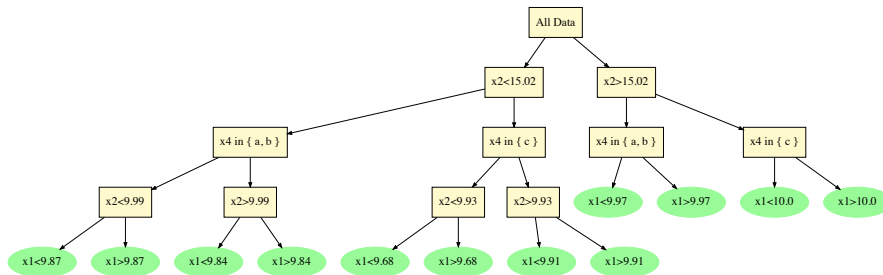

Figure 1: The optimally pruned tree $T(\alpha^*)$ of $\alpha^* = 0.0957$, with $|\widetilde{T}(\alpha^*)| = 12$ and $I(T(\alpha^*)) = 1.13$.

Furthermore, 100 repetitions of the simulation from (6) were made. Figure 2 provides the histograms of the estimated split levels on $X_1$, $X_2$ and $X_4$, collected from the 100 optimally pruned trees.

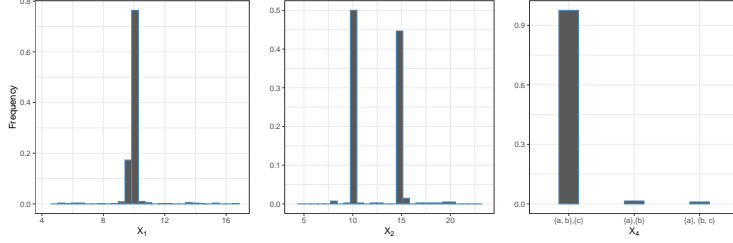

Figure 2: The histogram of split levels in 100 pruned trees

The selected splits concentrated around the genuine split levels with high probability. Specifically, 95% of the splits on $X_1$ were within $(9, 11)$ (the true split at $X_1 = 10$), 96% of the splits on $X_2$ were within $(9, 11) \cup (14, 16)$ (the true splits at $X_2 = 10$ and $15$), and nearly 98% of the splits on $X_4$ were in the form of $\{\{a, b\}, \{c\}\}$, This strongly supported the consistency results of the split selection procedure and validated the pruning procedures could effectively remove the redundant splits.

The second example is a regression function that does not conform the segmented linear form,

$$m(X) = \max\{e^{-10X_1^2}, e^{-50X_2^2} 1.25 e^{-5(X_1^2 + X_2^2)}\},$$

which was also considered in [22]. Figure 3 demonstrates the surface of $m(X)$ within the domain of $[-1, 1]^2$. We generated the training data of 1000 records $Y = m(X) + \varepsilon$, for $X_1, X_2 \overset{i.i.d}{\sim} U(-1, 1)$ and $\varepsilon \sim N(0, 0.01)$. With the same stopping parameter of $N_{\min} = 10, \text{Dep}_{\max} = 10$, we applied SLRT and CART respectively, obtaining the approximated surface in Figure 4 and 5.

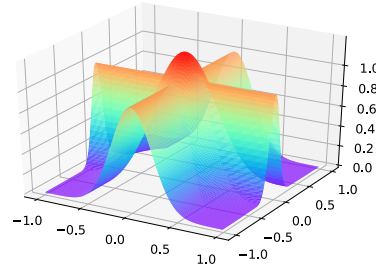

Figure 3: The true surface defined by $m(X)$

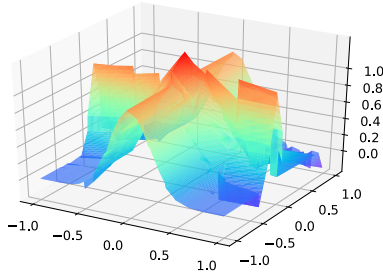

Figure 4: The approximated surface by SLRT

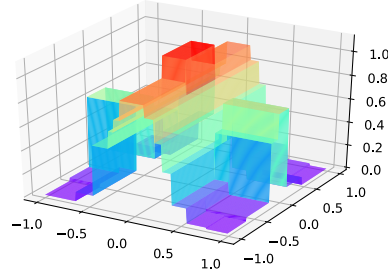

Figure 5: The approximated surface by CART

Then we calculated the root mean squared prediction errors (RMSPE) on an independent testing sample of size $500$. The RMSPE of SLRT is $0.047$, and that of CART is higher at $0.073$. Under this situation, SLRT obtains a locally linear approximation with a large tree structure, which tend to outperforms CART since it is locally constant. In practice, since the model complexity (the tree size) is adaptive to the nature of data, it depends on data whether the estimated regression function is a interpretable segmented linear approximation or a locally linear approximation.

## 5.2 Comparisons on Benchmark Datasets

The predictive performance is tested on $9$ benchmark datasets from the StatLib library [23] and the UCI Machine Learning Repository [24], where the sample sizes range from 74(Pyrimidine) to 39644(News Popularity) and 5 datasets include categorical variables. Detailed information about the covariates and sample sizes are reported in the supplementary materials.

The proposed SLRT with the least square estimation (SLRT$_{\text{LS}}$) and the LASSO (SLRT$_{\text{LASSO}}$) were compared with three tree-based methods: CART [11], GUIDE [17, 25] and MARS [26], with the

same $N_{\min}$ and $\text{Dep}_{\max}$ for all the methods. Ensemble predictors RF$_{\text{SLRT}}$ and WRF$_{\text{SLRT}}$ are random forests (RF) and the newly proposed weighted random forests (WRF) equipped with SLRT as the base predictor. The conventional RF based on CART (RF$_{\text{CART}}$) as well as WRF with CART (WRF$_{\text{CART}}$) were also implemented to serve as benchmarks. To make the results of RF and WRF comparable, their predictions were based on the same ensembles of 50 trees and only different in the way of aggregation. To evaluate the predictive performance, we divided each dataset into 10 subsets and implemented each method for 10 times using each subset as the testing set and the rest as the training set, where all methods shared the same training and testing sets. Table 1 summarizes the average RMSPEs from the 10-fold cross validation, where the integers in parentheses indicate the ranks within the single and the ensemble predictors, respectively.

Table 1: RMSPE (rank) of 10-fold cross-validation on 9 data sets: data name(sample size). The best performance is marked in ***bold italic***, within each group of single predictors and ensemble predictors.

| Datasets | Single Predictors | | | | | Ensemble Predictors | | | |
|---|---|---|---|---|---|---|---|---|---|
| | SLRT$_{\text{LS}}$ | SLRT$_{\text{LASSO}}$ | GUIDE | CART | MARS | RF$_{\text{SLRT}}$ | WRF$_{\text{SLRT}}$ | RF$_{\text{CART}}$ | WRF$_{\text{CART}}$ |
| Boston Housing(506) | 0.174(2) | ***0.170***(1) | 0.187(4) | 0.262(5) | 0.179(3) | 0.162(2) | ***0.158***(1) | 0.218(4) | 0.200(3) |
| ComputerHardware(209) | 47.89(2) | ***47.40***(1) | 48.06(3) | 62.60(5) | 54.17(4) | 38.49(2) | ***36.77***(1) | 64.91(4) | 39.08(3) |
| Auto-mpg(392) | 2.831(2) | ***2.791***(1) | 3.545(4) | 3.680(5) | 2.942(3) | 2.633(2) | ***2.614***(1) | 3.273(3) | 3.240(4) |
| Auto-mobile(159) | 0.154(2) | ***0.140***(1) | 0.231(5) | 0.192(4) | 0.184(3) | 0.143(2) | ***0.142***(1) | 0.165(4) | 0.162(3) |
| Kinematics(8192) | 0.139(2) | ***0.138***(1) | 0.140(3) | 0.257(5) | 0.198(4) | 0.117(2) | ***0.115***(1) | 0.249(4) | 0.247(3) |
| Abalone(4176) | 2.162(3) | ***2.143***(1) | 2.151(2) | 2.497(5) | 2.161(3) | 2.116(2) | ***2.113***(1) | 2.458(4) | 2.456(3) |
| Parkinson(5875) | 9.374(3) | 9.327(2) | ***9.300***(1) | 10.534(5) | 9.660(4) | 8.691(2) | ***8.679***(1) | 10.326(4) | 10.317(3) |
| Pyrimidine(74) | 0.088(2) | 0.093(3) | 0.096(5) | 0.094(4) | ***0.074***(1) | 0.078(4) | 0.076(3) | 0.073(2) | ***0.049***(1) |
| NewsPopularity(39644) | 0.877(4) | 0.872(2) | 0.873(3) | 0.903(5) | ***0.865***(1) | ***0.868***(1) | ***0.868***(1) | 0.901(3) | 0.901(3) |

Among the five single tree predictors, SLRT$_{\text{LASSO}}$ attained the best prediction in six dataset, MARS in two, and SLRT$_{\text{LS}}$ and GUIDE in one, respectively. This demonstrates the advantages of the proposed SLRT. Directly comparing SLRT with GUIDE, SLRT$_{\text{LS}}$ had better prediction in 7 out of the 9 datasets and SLRT$_{\text{LASSO}}$ in 8 out of 9 datasets. SLRT also compared favorably to MARS, having better performance in 6 out of the 9 datasets. CART appeared to be the worst predictor in seven datasets, while GUIDE ranked the last on the other two datasets. The better performance of SLRT$_{\text{LASSO}}$ over SLRT$_{\text{LS}}$ shows the benefits of conducting the variables selection on the leaf nodes.

The ensemble predictors RF$_{\text{SLRT}}$ and WRF$_{\text{SLRT}}$ showed better performance than the conventional RF with CART in 8 out of 9 datasets. Meanwhile, the ensemble predictors tended to outperform the single predictors, which suggests the effects of the bagging operation. The proposed WRF also showed improved predictions over the RF, which benefit from the weighting procedure that reduces the importance of those under-performing trees.

## 6  Conclusion

We propose a tree based approach called segmented linear regression trees (SLRT), which is based on two consecutive algorithms for partitioning structure learning: one for the split selection at each internal node based on a cumulative Kendall's $\tau$ statistic; the other for the parsimonious partitioning structure by tree pruning through an adaptive cost-complexity measure. Theoretical analysis shows that the split selection algorithm leads to the consistent identification and estimation of both the genuine split variables and the split levels, and the pruning procedure ensures the consistent estimation of the genuine number of segments. We implant the SLRT as the base predictor in RF and WRF to create two breeds of ensemble predictors. The proposed procedures are evaluated by numerical simulations and case studies, which shows advantageous predictive accuracy over other tree-based methods, and in creating more powerful breeds of ensemble predictors.

**Acknowledgments**

This research is funded by China's National Key Research Special Program Grant 2016YFC0207701, National Key Basic Research Program Grant 2015CB856000 and National Natural Science Foundation of China grant 71532001.

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
