[Supplementary Material]

# Partitioning Structure Learning for Segmented Linear Regression Trees (Supplementary Materials)

Xiangyu Zheng[1] and Song Xi Chen[1,2]
[1]Guanghua School of Management, Peking University
[2]Center for Statistical Science, Peking University

## Contents

## 1 Supplements for Section 1

See `https://github.com/xy-zheng/Segmented-Linear-Regression-Tree` for the code of the algorithm.

## 2 Supplements for Section 2

### 2.1 Lemma 2.1

**Lemma 2.1.** *Suppose Assumptions 2.1 through 2.4 hold. Let $(\hat{j}, \hat{a}) = \operatorname{argmax} C(j, a)$ and $\bar{C}(j, a)$ be the probability limit of $C(j, a)$. Then, for any $(j', a') \notin \mathcal{S}$, $\bar{C}(j', a') < \bar{C}(j_q, a_q)$ for any $(j_q, a_q) \in \mathcal{S}$.*

**Assumption 2.1.** $\mathrm{Var}(X|X \in D_l)$ *is diagonal for* $l \in \{1, \cdots, L_0\}$, *i.e. explicit regressors are uncorrelated conditional on each true segment.*

**Assumption 2.2.** $\mathrm{E}(\varepsilon|X) = 0$ *and* $\mathrm{Median}(\varepsilon|X) = 0$.

**Assumption 2.3.** $\mathrm{Median}\left(X_j - \mathrm{E}(X_j)|X_k\right) = 0$ *for* $1 \le j, k \le p$ *and* $k \neq j$.

**Assumption 2.4.** *For* $(j_k, a_k) \in \mathcal{S}$, *if* $(j', a') \neq (j_k, a_k)$, $0 < \mathrm{P}(X_{j_k} \le a_k | X_{j'} \le a') < 1$.

Assumption 2.1 assumes the conditionally uncorrelated regressors. Assumption 2.2 is satisfied when the conditional distribution of the residuals is symmetric and zero-mean, for instance, an independent Gaussian distributed noise. Assumptions 2.1 and 2.3 are satisfied under the stronger condition that $\{X_j\}_{j=1}^{p}$ are mutually independent and symmetrically distributed. Assumption 2.4 is for the identification of the true split. If $\mathrm{P}(X_{j_k} \le a_k | X_{j'} \le a') = 1$ for $(j', a') \neq (j_k, a_k)$, then the condition $X_{j_k} \le a_k$ is equivalent to $X_{j'} \le a'$ with probability 1. When $\mathrm{P}(X_{j_k} \le a_k | X_{j'} \le a') = 0$, then the condition $X_{j_k} \le a_k$ is equivalent to $X_{j'} > a'$. In either situation, we can not distinguish the split level $X_{j'} = a'$ from $X_{j_k} = a_k$, creating ambiguity in the identification of splits.

*Proof.* The segmented linear regression model is equivalent to

$$Y = \sum_{q=1}^{Q_0} \left\{ (\alpha_L^{(q)} + X'\beta_L^{(q)})I(X_{j_q} \le a_q) + (\alpha_R^{(q)} + X'\beta_R^{(q)})I(X_{j_q} > a_q) \right\} + \varepsilon, \qquad (2.1)$$

Firstly, consider the simplest case when $Q_0 = 1$ (In model (1), $L_0 = 2$),

$$Y = (\alpha_L + X^T\beta_L)\mathrm{I}(X_{j^*} \le a^*) + (\alpha_R + X^T\beta_R)\mathrm{I}(X_{j^*} > a^*) + \varepsilon, \qquad (2.2)$$

where both $j^*$ and $a^*$ are unknown. Note that the OLS estimation of slope parameters $\beta$ between $Y$ and $X$ with intercept is equivalent to the one between $Y - E(Y)$ and $X - E(X)$, and they also lead to the same residuals. Besides, the constant shift will not alter the threshold $(j^*, a^*)$. Therefore, without loss of generality, we assume $E(Y) = E(X) = 0$ and eliminate the intercept term.

Let $\boldsymbol{X} = (X(1), \cdots, X(n))^T$, $\boldsymbol{Y} = (Y(1), \cdots, Y(n))^T$, which are a matrix and a vector respectively. In addition, let $\boldsymbol{X}_L$ and $\boldsymbol{X}_R$ be defined as

$$\boldsymbol{X}_L = \mathrm{diag}\{I_{X_{j^*}(1) \le a^*}, \cdots, I_{X_{j^*}(n) \le a^*}\}\boldsymbol{X}, \ \boldsymbol{X}_R = \mathrm{diag}\{I_{X_{j^*}(1) > a^*}, \cdots, I_{X_{j^*}(n) > a^*}\}\boldsymbol{X}. \quad (2.3)$$

Then we have $\boldsymbol{X} = \boldsymbol{X}_L + \boldsymbol{X}_R$, $\boldsymbol{X}_L^T \boldsymbol{X}_R = \boldsymbol{X}_R^T \boldsymbol{X}_L = 0$. The OLS estimation is

$$
\begin{aligned}
\hat{\beta} &= (\boldsymbol{X}^T\boldsymbol{X})^{-1}\boldsymbol{X}^T Y \\
&= (\boldsymbol{X}^T\boldsymbol{X})^{-1}\boldsymbol{X}_L^T\boldsymbol{X}_L \cdot \beta_L + (\boldsymbol{X}^T\boldsymbol{X})^{-1}\boldsymbol{X}_R^T\boldsymbol{X}_R \cdot \beta_R + (\boldsymbol{X}^T\boldsymbol{X})^{-1}\boldsymbol{X}\varepsilon \qquad (2.4) \\
&= \frac{n_L}{n}(\frac{\boldsymbol{X}^T\boldsymbol{X}}{n})^{-1}\frac{\boldsymbol{X}_L^T\boldsymbol{X}_L}{n_L} \cdot \beta_L + \frac{n_R}{n}(\frac{\boldsymbol{X}^T\boldsymbol{X}}{n})^{-1}\frac{\boldsymbol{X}_R^T\boldsymbol{X}_R}{n_R} \cdot \beta_R + (\frac{\boldsymbol{X}^T\boldsymbol{X}}{n})^{-1}\frac{\boldsymbol{X}^T\varepsilon}{n}.
\end{aligned}
$$

Let $\Sigma = Var(X), \Sigma_L = Var(X|X_{j^*} \leq a^*), \Sigma_R = Var(X|X_{j^*} > a^*)$ and $\mu_L = E(X|X_{j^*} \leq a^*), \mu_R = E(X|X_{j^*} > a^*)$. According to Assumption 2.1, $\Sigma, \Sigma_L, \Sigma_R$ are all diagonal. For $E(X) = 0$, only the $j^*_{\text{th}}$ element of $\mu_L$ and $\mu_R$ are nonzero. Since $X(1) \cdots X(n) \overset{i.i.d}{\sim} X$ and $E(\varepsilon|X) = 0$ by Assumption 2.2, further we have

$$
\begin{aligned}
\hat{\beta} &= \frac{n_L}{n}\left(\Sigma + o_p(1)\right)^{-1}\left(\mu_L\mu_L^T + \Sigma_L + o_p(1)\right) \cdot \beta_L + \\
&\qquad\qquad \frac{n_R}{n}\left(\Sigma + o_p(1)\right)^{-1}\left(\mu_R\mu_R^T + \Sigma_R + o_p(1)\right) \cdot \beta_R + o_p(1) \\
&= \left(W \cdot \beta_L + (I_p - W) \cdot \beta_R + o_p(1)\right), \tag{2.5}
\end{aligned}
$$

as $n, n_L, n_R$ go to infinity. The weighting matrix $W = \mathrm{diag}\{w_1, \cdots, w_p\}$ is diagonal with positive elements ranges in $(0,1)$, where $w_j = P(X_{j^*} \leq a^*)\frac{E(X_j^2|X_{j^*}<a^*)}{Var(x_j)}$. Thus, $(\hat{\beta} - \beta_L)(\hat{\beta} - \beta_R) < 0$ when $n$ is sufficient large.

$$
\hat{e} = Y - X\hat{\beta} = (\alpha_L + X\tilde{\beta}_L)\mathrm{I}(X_{j^*} \leq a^*) + (\alpha_R + X\tilde{\beta}_R)\mathrm{I}(X_{j^*} > a^*) + \varepsilon + o_p(1),
$$

where $\tilde{\beta}_L = (I_p - W)(\beta_L - \beta_R)$ and $\tilde{\beta}_R = (-W)(\beta_L - \beta_R)$. Since $0 < w_j < 1$, we have $\tilde{\beta}_L(j)\tilde{\beta}_R(j) < 0$. Consider the marginal relation between $\hat{e}$ and $X_k$, $\hat{e}$ can be formulated as

$$
\begin{aligned}
\hat{e} &= \left(\alpha_L + b_{kL}X_j + \eta_{kL}\right)I(X_{j^*} \leq a^*) + \left(\alpha_R + b_{kR}X_j + \eta_{kR}\right)I(X_{j^*} > a^*) + o_p(1) \\
&\triangleq e + o_p(1),
\end{aligned}
$$

where $\eta_{kL} = \sum_{i \neq k} \tilde{\beta}_L(i)X_i I(X_{j^*} \leq a^*) + \varepsilon_j$, $\eta_{kR} = \sum_{i \neq k} \tilde{\beta}_R(i)X_i I(X_{j^*} > a^*) + \varepsilon_j$, $b_{kL} = \tilde{\beta}_L(k)$, $b_{kR} = \tilde{\beta}_R(k)$, and $e$ denote the probability limit of $\hat{e}$.

Now we examine the population version of Kendall's $\tau$ coefficient between $e$ and $X_k$, conditioning on a possible threshold $(j, a)$. For brevity, we omit the subscripts $k$ of $b_{kL}, b_{kR}, \eta_{kL}, \eta_{kL}$ in the following. When $(j, a) = (j^*, a^*)$,

$$
\begin{aligned}
\tau\left(e, X_k \Big| X_{j^*} \leq a^*\right) &= 2P\left((e - e')(X_k - X_k') > 0 \Big| X_{j^*} \leq a^*\right) - 1 \\
&= 2P\left(b_L(X_k - X_k')^2 + (\eta_L - \eta_L')(X_k - X_k') > 0\right) - 1 \\
&= 2P\left(\eta_L - \eta_L' > -b_L|X_k - X_k'|\right) - 1 \\
&\triangleq 2P\left(\tilde{\eta}_L < b_L|X_k - X_k'|\right) - 1, \tag{2.6}
\end{aligned}
$$

where $e'$ and $X_k'$ are independently and identically distributed as $e$ and $X$, $\tilde{\eta}_L := \eta_L' - \eta_L$. Similarly, we have

$$
\tau\left(e, X_k \Big| X_{j^*} > a^*\right) = 2P\left(\tilde{\eta}_R < b_R|X_k - X_k'|\right) - 1 \tag{2.7}
$$

Let $\tau_{k,L} := \tau\left(e, X_k \Big| X_{j^*} \leq a^*\right)$ and $\tau_{k,R} := \tau\left(e, X_k \Big| X_{j^*} > a^*\right)$. Due to Assumption 2.2 and 2.3, we have $\mathrm{Median}(\tilde{\eta}_L|X_j) = \mathrm{Median}(\tilde{\eta}_R|X_j) = 0$. Note that $b_L \cdot b_R < 0$, we have $\tau_{k,L} \cdot \tau_{k,R} < 0$ from (2.6) and (2.7).

For general $j$ and $a$, we have

$$\tau\left(e, X_k\Big| X_j \le a\right) \quad = \quad P(X_{j^*} \le a^* | X_j \le a) \cdot \tau_{k,L} + P(X_{j^*} > a^* | X_j \le a) \cdot \tau_{k,R}$$

$$\triangleq \quad p_{j,a} \cdot \tau_{k,L} + (1 - p_{j,a}) \cdot \tau_{k,R},$$

where $p_{j,a} = P(X_{j^*} \le a^* | X_j \le a)$. Since $\tau_{k,L} \cdot \tau_{k,R} < 0$ and $0 \le p_{j,a} \le 1$,

$$|p_{j,a} \cdot \tau_{k,L} + (1 - p_{j,a}) \cdot \tau_{k,R}| \le \min\left\{|p_{j,a} \cdot \tau_{k,L}|, |(1 - p_{j,a})\tau_{k,R}|\right\} \le \min\left\{|\tau_{k,L}|, |\tau_{k,R}|\right\}.$$

The arguments for $\tau(e, X_k \big| X_d > a)$ are similar. Let $q_{j,a} = P(X_{j^*} \le a^* | X_j > a)$, then

$$\left|\tau(e, X_k\big| X_d > a)\right| \quad = \quad |q_{j,a} \cdot \tau_{k,L} + (1 - q_{j,a}) \cdot \tau_{k,R}| \le \min\left\{|\tau_{k,L}|, |\tau_{k,R}|\right\} \quad .$$

Finally, we conclude that $\forall (j, a)$,

$$\left|\tau\left(e, X_k | X_{j^*} \le a^*\right)\right| - \left|\tau\left(e, X_k | X_j \le a\right)\right| = |\tau_{k,L}| - |p_{j,a} \cdot \tau_{k,L} + (1 - p_{j,a}) \cdot \tau_{k,R}| \ge 0, \quad (2.8)$$

$$\left|\tau\left(e, X_k | X_{j^*} > a^*\right)\right| - \left|\tau\left(e, X_k | X_j > a\right)\right| = |\tau_{k,R}| - |q_{j,a} \cdot \tau_{k,L} + (1 - q_{j,a}) \cdot \tau_{k,R}| \ge 0. \quad (2.9)$$

Then the conclusion follows directly from the definition in (**??**).

As for the general case of $Q_0 > 1$ ($Q_0$ in (2.1)), we firstly focus on one of the split thresholds, say $(j_1, a_1)$, and rewrite $Y$ as

$$Y = \sum_{l=1}^{2^{(Q_0-1)}} \left[(\alpha_L^{(l)} + X\beta_L^{(l)})\mathrm{I}(X_{j_1} \le a_1) + (\alpha_R^{(l)} + X\beta_R^{(l)})\mathrm{I}(X_{j_1} > a_1)\right] \mathrm{I}(X \in D_l^{(-1)}).$$

Here $D_l^{(-1)}$ is a intersection generated from $\mathcal{S}^{(-1)} = \{(d_q, a_q)\}_{q=2}^{Q_0}$, which maybe empty. As Kendall's $\tau$ coefficient is actually an expectation, we can apply the same procedures as discussed conditional on each $D_l^{(-1)}$, and use the law of iterated expectation to obtain the final result. Besides, note that being restricted in $D_l^{(-1)}$ means leaving out the other $(Q_0 - 1)$ split thresholds, we conclude that

$$\forall (j', a') \ne (j_q, a_q)_{q=1, \cdots, Q_0}, \ \bar{C}(j_1, a_1) > \bar{C}(j', a'), \quad (2.10)$$

Then applying the above deduction to any pair in $\mathcal{S} = \{(j_q, a_1)\}$, we conclude that $\forall (j', a') \notin \mathcal{S}$,

$$\bar{C}(j', a') < \bar{C}(j_q, a_q) \quad \text{for any} (j_q, a_q) \in \mathcal{S}. \quad (2.11)$$

□

## 2.2  Supplements for Theorem 2.1

*Proof.* Recall the definition in equation (2),

$$C(j, a) = \sum_{k=1}^{p} \left|\hat{\tau}\left(\hat{e}, X_k | X_j \le a\right)\right| + \left|\hat{\tau}\left(\hat{e}, X_k | X_j > a\right)\right|.$$

Since $\{X(i), Y(i)\}_{i=1}^{n}$ are independent, the sample version of Kendall's $\tau$ coefficient converges to its population version,

$$C(j, a) \xrightarrow{\mathrm{P}} \bar{C}(j, a), \text{ where } \bar{C}(j, a) := \sum_{k=1}^{p} \left| \tau\left(e, X_k | X_j \leq a\right) \right| + \left| \tau\left(e, X_k | X_j > a\right) \right|,$$

whose existence is assured by the law of large numbers. The population version of Kendall's $\tau$ coefficient is $\tau(X_k, \varepsilon | X_j \leq a) = 2\,\mathrm{P}((X_k - X_k')(\varepsilon - \varepsilon') > 0 | X_j \leq a, X_j' \leq a) - 1$, with $(X_k', \varepsilon')$ being an independent copy of $(X_k, \varepsilon)$.

Combing with (2.11), if $(j^*, a^*)$ is the unique solution to the population minimization problem $\min_{j,a} \bar{C}(j, a)$. Let $(\hat{j}, \hat{a})$ be the estimated split threshold,

$$(\hat{j}, \hat{a}) = \mathrm{argmax}_{j,a}\, C(j, a).$$

Therefore $(\hat{j}, \hat{a})$ belongs to a class of M-estimators, of which the consistency is guaranteed by Theorem 5.14 in Van der Vaart (2000).

For the general case where the maximum may not be unique, let $\Theta_0 := \mathrm{argmax}_{j,a} \bar{C}(j, a)$. By Theorem 5.14 in Van der Vaart (2000), for every $\varepsilon > 0$, $\mathrm{P}\left(d\left(\Theta_0, (\hat{j}, \hat{a})\right) > \varepsilon\right) \to 0$ as sample size $n$ goes infinity.

Furthermore, note (2.10) implies $\Theta_0 \subseteq \mathcal{S}$, thus we have $\mathrm{P}\left(d\left(\mathcal{S}, (\hat{j}, \hat{a})\right) > \varepsilon\right)$ $\to 0$ as $n$ goes to infinity. Especially, when there is a unique solution for $\max \bar{C}(j, a)$, denoted by $(j^*, a^*)$, then $(\hat{j}, \hat{a}) \xrightarrow{\mathrm{P}} (j^*, a^*)$. $\qquad\square$

## 2.3 Supplements for the transformation matrix $P^{(j,a)}$

The calculation of $P^{j,a}$ relies on the following proposition.

**Proposition 2.1.** *Let $M$ and $A$ be two symmetric square matrices of the same dimension, and $M$ be positive definite. Then there exists a non-singular matrix $P$ such that*

$$P^T M P = I \text{ and } P^T A P = \Lambda$$

*where $I$ is the identity matrix and $\Lambda$ is a diagonal matrix.*

*Proof.* Since $M$ is positive definite, we have the following decomposition

$$M = R^T R$$

for some non-singular matrix $R$. Then the matrix

$$\left(R^{-1}\right)^T A R^{-1}$$

is symmetric, so there exists an orthogonal matrix $B$ such that

$$B^{-1} \left(R^{-1}\right)^T A R^{-1} B = \Lambda$$

Set
$$P = R^{-1}B$$

Then $P^T = B^T \left(R^{-1}\right)^T = B^{-1} \left(R^{-1}\right)^T$, where we used that $B$ is orthogonal. So $P^T A P = \Lambda$. For the second equation, we use $B^T = B^{-1}$ and $\left(R^{-1}\right)^T = \left(R^T\right)^{-1}$ and obtain $P^T M P = B^{-1} \left(R^T\right)^{-1} R^T R R^{-1} B = I$. ☐

Recall that $\mathbf{X}_L = \mathbf{X} \operatorname{diag}\{\mathbb{I}(X_j(1) \leq a), \cdots, \mathbb{I}(X_j(n) \leq a)\}$. Applying Proposition 2.1 with $M = \mathbf{X}'\mathbf{X}$ and $A = \mathbf{X}'_L \mathbf{X}_L$, then $P$ is the transformation matrix $P^{(j,a)}$.

## 2.4 Supplements for Lemma 2.2 and Theorem 2.2

The difference between lemma 2.2 and lemma 2.1 is that lemma 2.2 does not require the covariates $X$ being uncorrelated. In the following, we provide the ideas of additional transformation. The detailed proof based on the transformed regressors would be similar to that of lemma 2.1. And the linkage between lemma 2.2 and theorem 2.2 is also Theorem 5.14 in Van der Vaart (2000).

Without loss of generality, consider the simplest case when $Q_0 = 1$,[1]

$$Y = (\alpha_L + X\beta_L)\mathrm{I}(X_{j^*} \leq a^*) + (\alpha_R + X\beta_R)\mathrm{I}(X_{j^*} > a^*) + \varepsilon, \tag{2.12}$$

Same as the equation (2.5) in Section 2.1, the OLS estimation is

$$
\begin{aligned}
\hat{\beta} &= (\boldsymbol{X}^T \boldsymbol{X})^{-1} \boldsymbol{X}^T Y \\
&= (\boldsymbol{X}^T \boldsymbol{X})^{-1} \boldsymbol{X}_L^T \boldsymbol{X}_L \cdot \beta_L + (\boldsymbol{X}^T \boldsymbol{X})^{-1} \boldsymbol{X}_R^T \boldsymbol{X}_R \cdot \beta_R + o_p(1).
\end{aligned}
$$

When $X$ is correlated on each partition, the probability limit of $(\boldsymbol{X}^T \boldsymbol{X})^{-1} \boldsymbol{X}_L^T \boldsymbol{X}_L$ and $(\boldsymbol{X}^T \boldsymbol{X})^{-1} \boldsymbol{X}_R^T \boldsymbol{X}_R$ are not diagonal. Hence we can not conclude that $\hat{\beta}(j)$ is between $\beta_L(j)$ and $\beta_R(j)$, which is the key in deriving the maximality of $|\tau|$ on the right split.

Note that $X^T X$ and $X_L^T X_L$ are both positive definite, they can be simultaneously diagonalized by congruence. There exist a nonsingular matrix $P$, such that both $P^T X^T X P$ and $P^T X_L^T X_L P$ are diagonal, which implies that $P^T X_R^T X_R P$ is also diagonal since $X^T X = X_L^T X_L + X_R^T X_R$.

Let $Z = XP, Z_L = X_L P, Z_R = X_R P$. Then we have $Z^T Z, Z_R^T Z, Z_L^T Z_L$ are all diagonal. Substituting $X = ZP^{-1}, X_L = Z_L P^{-1}, X_R = Z_R P^{-1}$ with into (2.12),

$$
\begin{aligned}
Y &= (\alpha_L + ZP^{-1}\beta_L)\mathrm{I}(X_{j^*} \leq a^*) + (\alpha_R + ZP^{-1}\beta_R)\mathrm{I}(X_{j^*} > a^*) + \varepsilon, \\
&\triangleq (\alpha_L + Z\theta_L)\mathrm{I}(X_{j^*} \leq a^*) + (\alpha_R + Z\theta_R)\mathrm{I}(X_{j^*} > a^*) + \varepsilon,
\end{aligned}
$$

where $\theta_L = P^{-1}\beta_L, \theta_R = P^{-1}\beta_R$. Applying the same procedure by calculating the Kendall's $\tau$ coefficients between $Y$ and $\{Z_j\}_{j=1}^p$, we can detect the optimal threshold of $X_{j^*} = a^*$.

# 3   Supplements for Section 3

## 3.1   Supplements for Proposition 3.1

**Proposition 3.1.** *Let $T_{\max}$ be the initial tree, then*

*(i)  given an $\alpha$, there exists one optimally pruned subtree $T(\alpha)$ of $T_{\max}$;*

*(ii)  if $\alpha_2 > \alpha_1$, then $T(\alpha_2)$ is a subtree of or equal to $T(\alpha_1)$.*

*Proof.* Consider a most simple binary tree with 3 nodes, $T_0 = \{t_0, t_L, t_R\}$. It has a trivial subtree is $T_1 = \{t_0\}$. From the definition of $I(t)$ and $I(T)$, we have

$$I_\alpha(T_0) = I(t_L) + I(t_R) + 2\alpha$$

and

$$I_\alpha(T_1) = I(t_0) + \alpha$$

It is straightforward to verify $I(t_0) \geq I(t_L) + I(t_R)$ from the definition of $I(t)$, since $I(t_0)$ is the sum of squared errors (SSE) of the restricted linear regression $(\beta_L = \beta_R)$ while $I(t_L) + I(t_R)$ is for the unrestricted equations.

Therefore, when $\alpha < I(t_0) - (I(t_L) + I(t_R))$, $T(\alpha) = T_0$. And when $\alpha \geq I(t_0) - (I(t_L) + I(t_R))$, $T(\alpha) = T_1$. Thus *(i)* and *(ii)* hold when $|T_0| = 3$.

Now suppose that *(i)* and *(ii)* are true when $|T_0| \leq k$, we only need to check they still hold when $|T_0| = k + 1$.

Let $T_L$ be the left branch of $T_0$ and $T_R$ the right one. It is easy to see,

$$\widetilde{T} = \widetilde{T_L} \cup \widetilde{T_R},$$

and

$$|\widetilde{T}| = |\widetilde{T_L}| + |\widetilde{T_R}|.$$

Therefore, $I_\alpha(T_0) = I_\alpha(T_L) + I_\alpha(T_R)$.

Then, $I_\alpha(T(\alpha)) = \min\{I_\alpha(t_0), I_\alpha(T_L(\alpha)) + I_\alpha(T_R(\alpha))\}$. If $I_\alpha(t_0) < I_\alpha(T_L(\alpha)) + I_\alpha(T_R(\alpha))$, $T(\alpha) = \{t_0\}$. Otherwise, $T(\alpha) = t_0 \cup T_L(\alpha) \cup T_R(\alpha)$. By induction, since $|T_L| \leq k$ and $|T_L| \leq k$, *(i)* and *(ii)* hold for $|T_0| = k + 1$ and further for all $T_0$.

□

## 3.2   Supplements for the convergence of $\widehat{L}$

The number of terminal nodes in the tree is equal to the number of segments of its derived segmented linear regression function. Starting with the overly grown tree $T_{\max}$ satisfying $|\widetilde{T}_{\max}| > L_0$, for $L_0$ being the number of the underlying partitions in Model (1), let $\widehat{L}$ be the number of leaf nodes in the optimally pruned subtree $T(\alpha^*)$. We consider the asymptotic property of $\widehat{L}$.

Let $\phi(u|x)$ be the conditional moment generating function of $\varepsilon$ given the explicit regressors $X$. To ensure the consistency of $\widehat{L}$, the conditional distribution of $\varepsilon$ is required to be locally exponentially bounded (LEB). Specifically, for

every compact set $B$ in the domain of $X$, there exist positive constants $u_0$ and $c_0$ such that
$$\phi(u|x) \leq e^{c_0 u^2} \text{ for any } |u| \leq u_0 \text{ and } x \in B.$$

The conditional LEB condition can be satisfied by $E(\varepsilon|X) = 0$ and $\phi(u|x)$ having bounded second derivative near zero for $x \in B$.

**Theorem 3.1.** *Suppose data is generated from model (1) with independent errors $\{\varepsilon(i)\}_{i=1}^n$ satisfying conditional LEB condition. The overly grown tree structure $T_{\max}$ has $\widehat{L}$ leaves and contains splits $\{(\hat{j}_q, \hat{a}_q)\}_{q=1}^{\widehat{Q}}$. Assume that*

(i) *$|\widetilde{T}_{\max}| > L_0$, where $|\widetilde{T}_{\max}|$ is the number of leaf nodes in $T_{\max}$;*

(ii) *For $(j_q, a_q) \in \mathcal{S}$, the set of underlying genuine splits, there exists a corresponding consistent estimate in $\{(\hat{j}_q, \hat{a}_q)\}_{q=1}^{\widehat{Q}}$.*

(iii) *Let $T_{true}$ be the tree structure that determines the underlying partitions $\{D_l\}_{l=1}^{L_0}$. Suppose $0 < \alpha^* < \min_{T \subseteq T_{true}, T \neq T_{true}} \left\{ \lim_{n \to \infty} I(T) - I(T_{true}) \right\}$.*

*Let $T(\alpha^*)$ be the optimal subtree of $T_{\max}$ by the minimal cost complexity pruning and $\widehat{L} = |\widetilde{T}(\alpha^*)|$. Then, $\widehat{L} \xrightarrow{P} L_0$ as $n \to \infty$.*

With sufficient large sample size, the first assumption can be satisfied by using smaller $N_{\min}$ and larger $\text{Dep}_{\max}$. The second assumption is a natural result of Theorem (2.1), which holds when Assumption 2.1 through 2.4 are satisfied. For the last assumption, since $(\alpha_{l_1}, \beta_{l_1}) \neq (\alpha_{l_2}, \beta_{l_2})$ for $l_1 \neq l_2$ in Model (1), we have $\lim_{n \to \infty} I(T) - I(T_{true}) > 0$ for $T \subseteq T_{true}, T \neq T_{true}$. Therefore, the upper limit of $\alpha^*$ is a positive constant, which guarantees the existence of a proper $\alpha^*$.

*Proof.* With out loss of generality, let us consider the leftmost branch. Suppose the leftmost branch in $T_{true}$ contains splits $\{X_{j_q} \leq a_q\}_{q=1}^{Q_1}$. Let $\hat{Q}_{1,\max}$ and $\hat{Q}_{1,\max}$ denote the splits contained in the leftmost branch of $T_{\max}$ and $T(\alpha^*)$. Then it suffices to prove $\hat{Q}_1 \xrightarrow{P} Q_1$. In this proof, let $D_0$ denote the training set in root node and $D_{Q_1}$ denote the data partition in $T_{true}$, that is, the subset determined by $\{X_{j_q} \leq a_q\}_{q=1}^{Q_1}$.

Let $\xi_q := (j_q, a_q), 1 \leq q \leq Q_1$ and $\hat{\xi}_q := (\hat{j}_q, \hat{a}_q), 1 \leq q \leq \hat{Q}_{1,\max}$. By assumption, we have $Q_{1,\max} > Q_1$ and $\hat{\xi}_q \xrightarrow{P} \xi_q$ for $q = 1, \cdots, Q_1$.

Let $\mathcal{D}(\xi_1, \cdots, \xi_Q)$ denote the set of data partitions determined by the $Q$ binary splits and $S_n(D)$ denotes the fitted sum of squares over domain $D$ by fitting $\{X(i), Y(i)\}_{X(i) \in D}$ in a linear regression function. Then let $S_n(\xi_1, \cdots, \xi_Q) := \sum_{D \in D_{Q_1} \cap \mathcal{D}(\xi_1, \cdots, \xi_Q)} S_n(D)$. The specific formulation of $S_n(D)$ is giver in 3.1 later.

Let $\boldsymbol{X_n} := (X(1), \cdots, X(n))^T$, $\boldsymbol{Y_n} := (Y(1), \cdots, Y(n))^T$, $\boldsymbol{\varepsilon_n} := (\varepsilon(1), \cdots, \varepsilon(n))^T$ and the matrix-formed indicator function $I_n(D) := \text{diag}(I_{X(1) \in D}, \cdots, I_{X(n) \in D})$,

followed by $X_n(D) := I_n(D)X$ and $Y_n(D) := I_n(D)Y$. Then when $D \in D_{Q_1} \cap \mathcal{D}(\xi_1, \cdots, \xi_Q)$ and $Q > Q_1$,

$$S_n(D) = Y_n^T[I_n(D) - H_n(D)]Y_n = \varepsilon_n^T[I_n(D) - H_n(D)]\varepsilon_n, \tag{3.1}$$

where $H_n(D) := X_n(D)[X_n(D)^T X_n(D)]^{-1} X_n(D)^T$, a positive definite matrix. Since $\mathcal{D}(\xi_1, \cdots, \xi_{Q_0})$ has limited elements less than $2^{Q_0}$ and $\hat{\xi}_q \xrightarrow{\text{P}} \xi_q$ for $1 \leq q \leq Q_0$, we have

$$S_n(\hat{\xi}_1, \cdots, \hat{\xi}_{Q_0})/n = S_n(\xi_1, \cdots, \xi_{Q_0})/n + o_p(1). \tag{3.2}$$

Combining (3.1) and (3.2) and by the definition of $S_n(\hat{\xi}_1, \cdots, \hat{\xi}_{Q_1})$,

$$S_n(\hat{\xi}_1, \cdots, \hat{\xi}_{Q_1})/n \leq \varepsilon_n^T \varepsilon_n / n + o_p(1). \tag{3.3}$$

As the finer partition will give no increase in the sum of squared fitting errors,

$$S_n(\xi_1, \cdots, \xi_{Q_1}, \hat{\xi}_1, \cdots, \hat{\xi}_{Q_{1,\max}}) \leq S_n(\hat{\xi}_1, \cdots, \hat{\xi}_{Q_{1,\max}}) \leq S_n(\hat{\xi}_1, \cdots, \hat{\xi}_{Q_1}). \tag{3.4}$$

By the formulation of (3.1) and the conditional LEB property of $\varepsilon$, we can concluded that $S_n(\xi_1, \cdots, \xi_{Q_0}, \hat{\xi}_1, \cdots, \hat{\xi}_{Q_{\max}}) = \varepsilon_n^T \varepsilon_n - O_p(\log(n)^2)$, which can be proved following exactly the same procedures in Lemma 5.1 and 5.2 of Liu et al. (1997). Then (3.4) leads to the following inequality,

$$S_n(\hat{\xi}_1, \cdots, \hat{\xi}_{Q_{1,\max}})/n \geq \varepsilon_n^T \varepsilon_n / n - o_p(1). \tag{3.5}$$

Combining the inequalities (3.2), (3.4) and (3.5), we obtain the key result,

$$S_n(\hat{\xi}_1, \cdots, \hat{\xi}_{Q_1})/n - S_n(\hat{\xi}_1, \cdots, \hat{\xi}_{Q_{1,\max}})/n = o_p(1). \tag{3.6}$$

Now we are prepared to investigate the property of $\hat{Q}$, the number of split thresholds contain in $T(\alpha^*)$. According to Section 3.2, $T(\alpha^*) = \operatorname{argmin}_{T \subset T_{\max}}\{I(T)/n + \alpha^*|\tilde{T}|\}$. Note that given $T_{\max}$,

$$\hat{Q}_1 = \underset{1 \leq Q \leq Q_{1,\max}}{\operatorname{argmin}} S_n(\hat{\xi}_1, \cdots, \hat{\xi}_Q)/n + \alpha^*(Q - Q_1).$$

Since $\alpha^*$ is a positive constant, we have $P(\hat{Q} \leq Q_0) \to 1$ from (3.6).

Now it suffices to prove $P(\hat{Q} \geq Q_1) \to 1$. As a matter of fact, $\forall Q < Q_1$, we have $S_n(\hat{\xi}_1, \cdots, \hat{\xi}_Q)/n = S_n(\xi_1, \cdots, \xi_{Q_1})/n + c + o_p(1)$ for some $c > 0$. The detailed proof can be found in Lemma 5.3 and 5.4 of Liu et al. (1997). Combining with (3.2), we have $P(\hat{Q} \geq Q_0) \to 1$ when $\alpha^* < c$. □

# 4   Supplements for Section 5

## 4.1   Supplements for the case of dependent regressors

For the case of dependent regressors, let $X_1 = U_1$, $X_2 = \frac{1}{2}X_1 + U_2$, $X_3 = \frac{1}{4}X_2 + U_3$, where $\{U_j\}_{j=1}^3$ are independently distributed as $U(0, 20)$, $U(0, 15)$ and $U(0, 5)$.

(a) the tree constructed by the testing-based stopping rules.

(b) the tree constructed by simple stopping rule and tree pruning

Figure 1: Tree Structures in the Case of Dependent Regressors.

The discrete variable $X_4$ is still independent with $\{X_j\}_{j=1}^3$, which was not an explicit regressor. The training data of 1500 observations was also generated from $Y = m(X) + \varepsilon$, where $\varepsilon \sim N(0, 1)$. Figure 1a shows the tree generated by the testing based stopping rule with significance level 0.05, which has 12 terminal nodes, equal to the number of underlying segments. All the splits in Figure 1a are also very close to the underlying ones. Figure 1b shows the optimally pruned tree $T(\alpha^*)$ with $\alpha^* = 0.0593$, which is the subtree of $T_{\max}$ grown by *Algorithm 2* with $N_{\min} = 40$ and $\text{Dep}_{\max} = 10$. There are 13 terminal nodes in the pruned tree, including one redundant split of $X_1 = 17.27$, which may result from the insufficient sample size of its parent node.

Table 1: Frequencies of selected split variables fell within certain ranges (specified in the table) using two stopping rules: simple stopping plus tree pruning and the stopping rule based on hypothesis testing (Dependent Regressors).

|  | Simple stopping + Tree pruning | Testing based stopping |
|---|---|---|
| the frequency of splits on $X_1$ that fell within $(9, 11)$ | 91.6% | 81.1% |
| the frequency of splits on $X_2$ within $(9, 11) \cup (14, 16)$ | 86.2% | 87.5% |
| the frequency of splits on $X_4$ that were $\{\{a, b\}, \{c\}\}$ | 99.6% | 100% |

Furthermore, we also replicated the above procedures for 100 times. Table 1 summarized the frequencies of selected splits around the genuine ones. The averaged number of terminal nodes are 11.17, 11.48, 24.31 for testing-based stopping, pruning, and simple stopping respectively. Figure 2 shows that in the case of dependent regressors, both pruning and the testing-based stopping rule also worked well in obtaining the right-sized trees while the simple stopping rule tend to generate an overly grown tree.

## 4.2 Supplements for the detailed information of the datasets

**Boston Housing**  The data concerns the median housing values (medv) on 506 census tracts around Boston, where 13 covariates are collected. We take the logarithm of medv as the response, to be consistent with Loh (2002).

**Computer Hardware**  The target variable is the published relative CPU performance. There are 6 numeric predictive attributes about the cycle time, memory and number of channels.

**Auto-mpg**  The target variable mpg refers to city-cycle fuel consumption in miles per gallon. We discards the 'car name' attribute and use the other 7 attributes for the prediction of mpg.

**Auto-mobile**  This dataset has 159 complete records of 26 attributes. We use

Figure 2: Density plot of the number of terminal nodes in the 100 replications of three types of trees: the tree by simple stopping, the tree by simple stopping plus pruning, the tree by the stopping rule based on hypothesis testing (Dependent Regressors).

Table 2: Datasets, the sample sizes, and the dimensions of predictive attributes.

| Datasets | Sample Size | Number of Predictive Attributes (Numeric+ Categorical) |
|---|---|---|
| Boston Housing | 506 | 13 (12+1) |
| Computer Hardware | 209 | 6 (6+0) |
| Auto-MPG | 392 | 7 (4+3) |
| Auto-mobile | 159 | 25 (15+10) |
| Pyrimidine | 74 | 26 (26+0) |
| Kinematics | 8192 | 8 (8+0) |
| Parkinson | 5875 | 16 (16+0) |
| Abalone | 4176 | 8 (7+1) |
| News Popularity | 39644 | 47 (44+3) |

the other 25 attributes to predict the logarithm of price, where the 'make' attribute is transformed into a binary variable.

**Pyrimidine** The goal is to explore the inhibition of dihydrofolate reductase (DHFR) by pyrimidines using the information of 74 pyrimidines. The predictive attributes include 26 numeric variables about the physiochemical and structural properties of pyrimidines.

**Kinematics** This dataset contains 8192 cases with 9 attributes. The goal is to predict the distance of the end-effector from the target, using $\theta_1, \cdots, \theta_8$, the 8 angular positions of the joints as predictive attributes.

**Abalone** The relevant task is to predict the age of abalone from the nominal variable sex and 7 the other numeric variables of physical measurements.

**Parkinson** The dataset is composed of a range of biomedical voice measurements from people with early-stage Parkinson disease (Little et al., 2009). We use the 16 voice measures to predict the total UPDRS.

**News Popularity** The dataset collects 58 features of articles published by Mashable in two years (Fernandes et al., 2015), including 39644 complete cases. Aggregating the 13 variables about 'weekday' and 'channel' into two categorical variables, we use the 47 covariants to predict the $\log$(shares).

## Footnotes

[1]Discussion for general case is the same with Section 2.1.