[Reviews · NeurIPS 2019]

Reviewer 1



This paper considers the piecewise linear regression problem and proposes a split selection algorithm followed by a pruning operation to capture the underlying partitions of the piecewise linear model. The language of the paper is okay but it is hard to follow the paper at certain points due to the heavy notation. I would recommend the authors to provide a verbal description of the meaning of such equations and inequalities. The main idea of the paper is interesting but I'm not sure if it is enough to be published at NeurIPS. Theoretical results do not really provide any intuition about how well the algorithm will perform when implemented, although I acknowledge that proving such a result is tricky. Given that the theoretical contribution of the paper is not outstanding, I would've expected the experiments to provide more insight about the algorithm. However, the proposed algorithm is only compared against 20-30 year old methods, which does not say much about its performance. I would recommend the authors to see the following more recent articles on tree-based algorithms: Wang et al. "Local Supervised Learning through Space Partitioning" 2012. Vanli et al. "A Comprehensive Approach to Universal Piecewise Nonlinear Regression Based on Trees" 2014. Ikonomvska et al. "Online tree-based ensembles and option trees for regression on evolving data streams" 2015. Based on the above reasons, I think this paper is not ready for publication yet. As a side note, the supplementary material contains huge amount of typos, so I would recommend the authors to go over it carefully.

Reviewer 2



Some comments per section: Section 2 - Line 128: I am not sure how the distance measure between two pairs (pair of index and split value) should be interpreted/computed. - In case of presence of correlated covariates, is the algorithm able to discover the required linear transformation simultaneously with the required local split value? Section 3 - As for correlated covariates, please comment on how this is handled in Algorithm 2. I don't see any reference of matrix $P$. Experiments - How big is the ensemble size? - Please comment on how correlated covariates were handled for each of the datasets. Summary Originality: the proposed method is an easy extension of existing methodologies. Quality: overall quality is average because of the clarity problem (see below). Clarity: the paper overall is clearly written - however, could use some work in organization and maybe bring in one of the theoretical results inside the main writeup. Significance: because the work is largely based on existing methodologies, either a strong theoretical result or an empirical result should have been provided. Given that the theoretical result in this paper is not quite strong, the paper is more empirically oriented.

Reviewer 3



The proposed partitioning process fit the least square regression over the node and select the optimal split. In addition, for each leaf node, Lasso linear regression model is fitted. My concern is that this can be very slow for large-scale datasets. It is interesting to see the empirical time cost in the experiments. In the experiments, authors compared baselines such as GUIDE, CART and MARs on 9 datasets, and found CART is the worst method. It is reasonable to think CART might not be the best for base classifiers in the ensemble. Although authors compared WRF of SLRT with RF/WRF of CART, it is better to have comparing results with base classifier such as GUIDE and MARS to see how much the proposed method can improve. LASSO linear regression on leaf nodes is used for prediction. How did authors determine the regularization parameter in the experiments? How sensitive it is to affect the predictive results.

Reviewer 4



Originality: The paper is fairly original in that it proposes a new tree-splitting criterion that seems to work very well when the leaves are linear models rather than constants. It also provides a novel application of several pieces of previous work, including LASSO and random forests. There are adequate citations of related work. Quality: I did not carefully check the math or read the proofs in the supplemental material, but I did not observe any technical mistakes. There is not much discussion of the limitations of their approach. Clarity: The paper is written very clearly; I could understand everything without difficulty. Significance: This paper does seem to advance the state of the art in learning with tree-based models. Unless their experimental section is misleading, I expect this work to be generally useful to researchers and practitioners of tree-based modeling. Detailed notes: * You mention that the SLRT algorithm is consistent; does this property hold only in cases where the response variable is truly a piecewise-linear function of the regressors, with axis-aligned rectangular subspaces? I would expect the majority of real-world data not to conform to this particular shape. What can be said about SLRT in such cases? * Line 38: What is meant by "prevailing" in this sentence? You mention theoretical analysis, so I would assume you have some precise definition in mind, but I don't know what it would be. * Line 46: This sentence seems ungrammatical. Did you mean "Futhermore, by implanting SLRT as the base predictor..."? * Line 76: Typo: "method" should be "methods". * Line 305: Typo: "SIsson" should be "Sisson". * In Equations (2) and (4), you include examples that are equal to the split level in the Left child, but in Line 8 of Algorithm 1, you include them in the Right child. Is this a mistake? * How do you determine whether to use Algorithm 1 or Algorithm 2? Do you use some threshold for the sample correlations between pairs of regressors? * Line 193: Did you mean "subset or equal to" instead of "strict subset"? * Line 202: You don't want to "minimize the average predictive accuracy", so maybe replace "minimize" with "optimize". * Line 203: Typo, replace "optimially" with "optimal". * Line 232: I think the \sigma_i in the numerator should be \sigma_b.

[Author Response · NeurIPS 2019]

We thank all reviewers for the valuable comments! It appears that all reviewers saw the algorithmic and theoretical
contributions of our work. Below we provide responses to the questions and concerns of the reviewers.

(**R1**) *Comparisons with more recent works on tree-based algorithms.* Thanks for your suggestions and the references!
*Qualitative comparisons:* As we mentioned in line 69, [1] dealt with classification (not regression) by space partitioning
and locally linear classifiers. [2] focused on the ensemble algorithms for *online regression*, which also built on trees
with linear models in leaves, but using existing tree construction algorithms. [3] introduced two tree-based methods for
*adaptive learning*, where DFT used fixed hard splits (not trained) and DAT used adaptive soft splits trained by stochastic
gradient descent (SGD). In comparison, we use adaptive hard splits trained by looking into the rank correlation between
the residuals and covariates, instead of directly minimizing the empirical error by SGD. [3] focused on the weighting in
averaging all piecewise functions defined by the subtrees, while we focused on the split selections in tree constructions.
*Experiments comparisons:* It turns out that [2] shared dataset 'Abalone' and [3] shared dataset 'Kinematics' with us.
Table 3 in [2] reported that the ORF was the best with $\text{MSE}=5.68$ by CV, while our Table 1 shows $\text{SLRT}_{\text{LASSO}}$ had
lower $\text{MSE}=4.562(2.136^2)$ and the ensemble version $\text{RF}_{\text{SLRT}}$ had even smaller $\text{MSE}=4.439(2.107^2)$. [3] reported
DAT was the best with time accumulated error 0.0639 on the normalized data. To be comparable, we re-normalized
'Kinematics' by $\frac{y-(\max(y)+\min(y)/2}{(\max(y)-\min(y))/2}$. The MSEs by the 10-fold CV were $0.0326(\text{SLRT}_{\text{LS}})$ and $0.0327(\text{SLRT}_{\text{LASSO}})$,
respectively, which were significantly lower than DAT. Nevertheless, we admit the results were not entirely comparable
since [3] calculated the time accumulated errors in the adaptive learning context while we used cross validation. On the
two common datasets of both articles, SLRT shows promising performance despite our motivation is not to minimize
the empirical error directly but to maximize the rank correlation between the residuals and covariates in splits selection.
We've emailed the authors of [2] and [3] for experiment details and codes, and will make fairer comparisons in the final.

(**R1 & R2**) *Theoretical contributions and assumptions.* Theoretical justifications in Sec 2.2 show the selected splits
converge in probability to the true splits, which motivated Alg1 (uncorrelated regressors) and Alg2 (correlated
regressors). Although the convergence is in the asymptotic sense, it does provide mathematical assurance for consistent
split selections, which is the force behind the promising performance in the simulations in Sec 5. Regarding assumptions
in the supplement, ASMP 2.2-2.4 are quite general with ASMP 2.3 satisfied for symmetrically distributed $X$ that can be
realized by Box-Cox transformation, and ASMP 2.4 for identification purpose. We agree ASMP 2.1 on the correlated
regressors is strong, which is the reason for providing theoretical results and Alg2 for the correlated regressors.

(**R1 & R2**) *Clarity issues.* We'll bring in the theoretical results in the main writeup, and add more explanations for the
analysis in Sec 2.2 with illustrations by the example in Sec 5.1. We'll update the supplement to make it issues-free.

(**R1 & R4**) *Piecewise linearity concerns/ Generalization of models at the leaves.* To accommodate non-linearity at the
leaves, higher order polynomial regressors can be added. This extension is essentially covered by the existing Alg2 for
correlated regressors, with consistency of splits selection ensured by Th 2.2. We'll add the discussion in the final paper.

(**R2**) *How the algorithm optimizes $P$.* Sorry for not presenting clearly. The transformation matrix $P_{(j,a)}$ in line 137 is
directly calculated, without optimization. Let $S_X^L$ and $S_X^R$ be the sample covariance matrices conditional on $X_j \leq a$ and
$X_j > a$, respectively. Then, $P_{(j,a)}$ satisfies: $P_{(j,a)}S_X^L P_{(j,a)}^T$ and $P_{(j,a)}S_X^R P_{(j,a)}^T$ are both diagonal matrices, which can be
obtained by the spectral decomposition of positive definite matrices. We will add more details in the final paper.

(**R2 & R3**) *Experiment settings.* The ensemble size in the experiments was 50. We conduct the linear transformation
on X by multiplying $(S_X^{1/2})^{-1}$ to remove the correlation, where $S_X$ is the sample covariance matrix. We'll add the
computational complexity analysis, RF and WRF ensembles with GUIDE and MARS as the base methods in the final
version. Information about responses and covariates on the nine datasets were introduced in Sec 5 of the supplement.

(**R2**) *Distance between pairs.* The distance in line 128 is the Euclidean norm, where index $j$ is regarded as an integer.

(**R3**) *How to determine the regularization parameter $\lambda$ in LASSO?* By the cross validation over each node.

(**R4**) *A Theoretical discussion of the case when the data does not conform to the SLR setting.* In such cases, the induced
tree would be large in size as the model complexity is adaptive to data, and the estimation tends to be a nonparametric
first-order approximation. The theoretical discussion would be similar to the results in [4]. So, when the data conform
to the SLR setting (approximately), SLRT provides interpretable results with a concise tree structure; otherwise, SLRT
also provides fine approximations using elaborate partitions by a large tree. We'll add the discussion in the final paper.

(**R4**) *Detailed comments.* *Line 38:* "prevailing" means attaining the maximum of $\bar{\mathcal{C}}(j,a)$. *Line 8 of Alg1:* Equations (2)
and (4) were right. Sorry for the typo in Alg1. *How to determine whether to use Alg1 or Alg2?* This can be judged by
testing on the covariance of $X$ being diagonal or not using the sphericity test. *Line 193:* "subset or equal to".

Minor Issues. All minor issues raised by the reviewers will be rectified in full. Thanks for all constructive suggestions!

## References

[1] J. Wang and V. Saligrama, "Local supervised learning through space partitioning," in *NeurIPS*, 2012, pp. 91–99.
[2] E. Ikonomovska *et al.*, "Online tree-based ensembles and option trees ... on evolving data streams," *Neurocomputing*, 2015.
[3] N. D. Vanli *et al.*, "A comprehensive approach ... regression based on trees," *IEEE Transactions On Signal Processing*, 2014.
[4] P. Chaudhuri *et al.*, "Piecewise-polynomial regression trees," *Statistica Sinica*, 1994.


[Meta-Review · NeurIPS 2019]

The paper proposes and investigates how to learn tree structure for linear regression trees based on a conditional Kendall’s tau statistics with theoretical analysis.The ideas were new and generally satisfying to reviewers. While some reviewers would have liked to see even more experiments and experimental comparisons and details, other reviewers felt that the author response about the experiments was satisfying.